# HowToCaption: Prompting LLMs to Transform Video Annotations at Scale

## Abstract

Instructional videos are an excellent source for learning multimodal representations by leveraging video-subtitle pairs extracted with automatic speech recognition systems (ASR) from the audio signal in the videos. However, in contrast to human-annotated captions, both speech and subtitles naturally differ from the visual content of the videos and thus provide only noisy supervision for multimodal learning. As a result, large-scale annotation-free web video training data remains sub-optimal for training text-video models. In this work, we propose to leverage the capability of large language models (LLMs) to obtain fine-grained video descriptions aligned with videos. Specifically, we prompt an LLM to create plausible video descriptions based on ASR narrations of the video for a large-scale instructional video dataset. To this end, we introduce a prompting method that is able to take into account a longer text of subtitles, allowing us to capture context beyond a single sentence. To align the captions to the video temporally, we prompt the LLM to generate timestamps for each produced caption based on the subtitles. In this way, we obtain human-style video captions at scale without human supervision. We apply our method to the subtitles of the HowTo100M dataset, creating a new large-scale dataset, HowToCaption. Our evaluation shows that the resulting captions not only significantly improve the performance over many different benchmark datasets for text-video retrieval but also lead to a disentangling of textual narration from the audio, boosting performance in text-video-audio tasks.

## 1 Introduction

Textual descriptions of visual information allow for navigating large amounts of visual data. Improving the alignment between visual and textual modalities is crucial for many applications, *e.g.*, in the context of text-video retrieval to identify videos based on the described content. Recently, image-text cross-modal learning has achieved remarkable performance (Radford et al., 2021) in many downstream tasks by pre-training on large-scale web datasets consisting of text-image pairs. To collect video data on a similar scale, media platforms such as YouTube can be used as a great source of freely available videos (Abu-El-Haija et al., 2016; Zhou et al., 2018). Most of these videos include some narrations, *e.g.*, in instructional videos (Miech et al., 2019), people explain and show how to accomplish one or another task. To transform spoken language from the videos into subtitles, current automatic speech recognition (ASR) systems (Radford et al., 2023) can be used, providing aligned text-video annotated pairs for free. This automatic supervisory signal can easily scale to large video datasets. However, such video web data poses additional challenges (Han et al., 2022; Miech et al., 2019): (1) spoken and visual information in the video can deviate from each other, *e.g.*, when speakers provide information beyond what is visible or when spoken instructions do not temporally align with the actions shown, (2) speech contains filler words and phrases, such as "I'm going to", and can be incomplete and sometimes contains grammatical errors, and (3) ASR transcripts usually do not have punctuation and may contain errors. Therefore, a simple matching of videos and corresponding ASR subtitles provides only weak, noisy information, and the learned representations are not as generalizable as similar image-text representations learned on web data (Miech et al., 2019).

To address this problem, we propose a new framework, HowToCaption, that leverages large language models (LLMs) (Chiang et al., 2023) to generate human-style captions on a large scale for web-video instructional datasets based on corresponding ASR subtitles. By carefully designing prompts, we show that the LLM can effectively map long, noisy subtitles into concise and descrip-

tive human-style video captions. This approach allows us to create a large-scale dataset of video captions without any human supervision. Moreover, we can obtain a temporal alignment between the generated captions and specific moments in the given video sequences by tasking the LLM to predict timestamps for each caption. Our method can generate aligned text-video pairs on a large scale without human intervention. For additional quality improvement, we apply filtering and realignment within short temporal windows with respect to the generated timestamp. Beyond providing better annotation, the new captions provide the advantage that they are no longer a direct output of the speech signal, thus effectively decoupling audio and text. Current methods usually avoid using audio (Miech et al., 2020; Han et al., 2022), as the ASR subtitle is directly derived from speech, thus leading to the problem that any text-to-audio+video retrieval would mainly retrieve the closest speech signal while disregarding the video. Being able to generate captions that deviate from the speech thus allows to extend retrieval to audio+video without the need for fine-tuned regularization.

To verify the effectiveness of the proposed HowToCaption method, we generate new captions for the large-scale HowTo100M dataset (Miech et al., 2019). We evaluate the quality of the improved narrations on various challenging zero-shot downstream tasks over four different datasets, namely YouCook2 (Zhou et al., 2018), MSR-VTT (Xu et al., 2016), MSVD (Chen & Dolan, 2011), and LSMDC (Rohrbach et al., 2015). It shows that the generated captions not only provide a better training signal but also allow for a decoupling of speech and caption annotation, allowing a retrieval based on audio, vision, and subtitles at scale. We release a new HowToCaption dataset with high-quality textual descriptions to show the potential of generated captions for web text-video pairs. We also make code publicly available.

We summarize the contributions of the paper as follows:

- We propose a HowToCaption method to efficiently convert noisy ASR subtitles of instructional videos into fine-grained video captions, which leverages recent advances in LLMs and generates high-quality video captions at scale without any human supervision.

- We create a new HowToCaption dataset with high-quality human-style textual descriptions with our proposed HowToCaption method.

- Utilizing the HowToCaption dataset for training text-video models allows us to significantly improve the performance over many benchmark datasets for text-to-video retrieval. Moreover, since new textual annotation allows us to disentangle audio and language modalities in instructional videos, where ASR subtitles were highly correlated to audio, we show a boost in text-video+audio retrieval performance.

## 2    RELATED WORK

As this work introduces a method to improve ASR-based Video-Language datasets, which is centered around LLM, we organize the related work into three categories: related datasets, learning from ASR data, and LLM in vision-language tasks.

### 2.1    LARGE-SCALE VIDEO-LANGUAGE DATASETS

Manual annotation of video captioning datasets is even more time-consuming than image captioning since it involves video trimming and localization of caption boundaries. Currently, manually annotated video captioning datasets, e.g., MSR-VTT (Xu et al., 2016), and YouCook2 (Zhou et al., 2018), are limited in size. Therefore, different methods of mining video with weak supervision from the Internet were considered. Such datasets as YouTube-8M (Abu-El-Haija et al., 2016) and IG-Kinetics-65M (Ghadiyaram et al., 2019) provided multiple class labels based on query click signals and metadata (Abu-El-Haija et al., 2016) or hashtags (Ghadiyaram et al., 2019). However, short class labels are a suboptimal supervision compared to textual descriptions (Desai & Johnson, 2021). Therefore, Bain et al. (2021) considered scrapping from the web videos with associated alt-text, similarly to image-based Conceptual Captions dataset (Sharma et al., 2018), obtaining the WebVid2M dataset (Bain et al., 2021) that contains 2.5M videos-text pairs. Stroud et al. (2020) proposed to use meta information, such as titles, video descriptions, and tags from YouTube, as a textual annotation and created the WTS-70M dataset. And Nagrani et al. (2022) proposed to transfer image captions from an image-text dataset to videos by searching videos with similar frames to the image and,

therefore, collected the VideoCC3M dataset. However, most videos in the WebVid2M dataset do not have audio, which is an essential part of video analysis, and captions in the VideoCC3M dataset are derived from images and, therefore, tend to describe more static scenes rather than actions. At the same time, the title and tags of WTS-70M provide only high-level video descriptions.

As an alternative to this, Miech et al. (2019) proposed the HowTo100M dataset, where instructional videos are naturally accompanied by dense textual supervision in the form of subtitles obtained from ASR (Automatic Speech Recognition) systems. The HowTo100M dataset with 137M clips sourced from 1.2M YouTube videos was proven to be effective for pre-training video-audio-language representations (Rouditchenko et al., 2021; Chen et al., 2021; Shvetsova et al., 2022). The followed-up YT-Temporal-180M (Zellers et al., 2021) and HD-VILA-100M (Xue et al., 2022) datasets are created by using the same idea, but expand the HowTo100M with a larger number of videos, higher diversity, and higher video resolution. While ASR supervision can provide a scalable way to create a large video dataset with dense annotation, the quality of ASR subtitles is still not on par with human-annotated captions. In this work, we propose a method to create high-quality captions for videos at scale by leveraging LLM and subtitles.

## 2.2 LEARNING WITH NOISY ASR SUBTITLES OF INSTRUCTIONAL VIDEOS

The problem of misalignment and noisiness of ASR supervision in instructional videos, such as in the HowTo100M dataset, were addressed in multiple works. MIL-NCE loss (Miech et al., 2020) and soft max-margin ranking loss (Amrani et al., 2021) were proposed to adapt contrastive loss to misalignment in text-video pairs. Zellers et al. (2021) proposed to use LLM to add punctuation and capitalization to ASR subtitles and remove mistranscription errors. Han et al. (2022) proposed to train temporal alignment networks to filter out subtitles that are not alignable to the video and determine alignment for the others. However, to the best of our knowledge, (Lin et al., 2022) is the only work that goes beyond just removing mistranscription errors and ASR re-alignment, where Lin et al. (2022) proposed to match the sentences from ASR subtitles to a large base of descriptions of the steps from wikiHow dataset (Koupaee & Wang, 2018) (distant supervision). In our work, we propose to use LLM to create video captions given ASR subtitles, which allows us to create a detailed description that is specific for every video and has proper sentence structure.

## 2.3 LARGE PRE-TRAINED LANGUAGE MODELS IN VISION-LANGUAGE TASKS

In recent years, there has been a remarkable success of LLMs in many language-related tasks (Devlin et al., 2018; Radford et al., 2019; Raffel et al., 2020). Latest large language models such as GPT-3.5 (Neelakantan et al., 2022), Alpaca (Taori et al., 2023) or Vicuna (Chiang et al., 2023) have demonstrated excellent zero-shot capabilities on common sense inference (Chang & Bergen, 2023). This success has prompted research into integrating common-sense knowledge into vision-language tasks to enhance their performance. In this regard, some methods (Sun et al., 2019; Su et al., 2019; Lu et al., 2019; Tan & Bansal, 2019) initialize the language part of vision-language models from pre-trained LLM. Another line of work (Cho et al., 2021; Li et al., 2023; Zhao et al., 2023) uses LLM as a decoder to enable vision-to-language generation. For example, the MiniGPT-4 (Zhu et al., 2023) model enhances a frozen Vicuna model by aligning visual encoder tokens with Vicuna's input token space, enabling visual reasoning capabilities, *e.g.*, image question answering or image captioning. In this regard, some works (Lialin et al., 2023; Zhao et al., 2023) adapted visually conditioned LLM for visual captioning and created captioning pseudo labels for large-scale video data that later used for vision-language tasks. However, these methods require human-annotated datasets to train a captioning model , while our method does not require any label data and aims to transform free available annotation (ASR subtitles) into textual descriptions.

## 3 METHOD

### 3.1 PROBLEM STATEMENT

Given a dataset of $N$ untrimmed long-term instructional videos $V_n$ with corresponding noisy ASR (automatic speech recognition) subtitles $S_n$, our goal is to create "human-like" fine-grained video captions $C_n$ (with $1 \leq n \leq N$). Note that the task does not assume access to any paired training

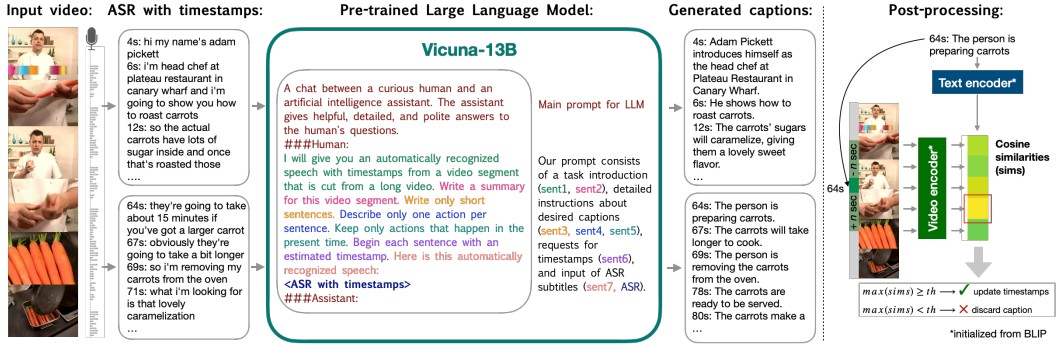

Figure 1: . **Schematic visualization the proposed HowToCaption method**. Obtained from Automatic Speech Recognition System (ASR) subtitles divided into blocks that contain longer contextual information. A large pre-trained language model is then used to generate plausible video captions based on ASR subtitles, along with timestamps for each caption. These generated captions and timestamps are further additionally post-processed to enhance their alignment to the video and filter out captions with low similarity to the corresponding video by leveraging a pre-trained text-video model.

data $((V_n, S_n), C_n)$. The goal is to create the video captions $C_n$ in a *zero-shot* setting given only videos and subtitles $(V_n, S_n)$. More formally, for each given video $V_n$, we also obtain a set of sentences that people have spoken in the video, $S_n = \{s_{n,j}, t^s_{n,j}, t^e_{n,j}\}_{j \leq |S_n|}$ with their start $t^s$ and end timestamps $t^e$ recognized by ASR-systems. Our goal is for each video $V_n$ to generate dense captions $C_n = \{c_{n,i}, \tau^s_{n,i}, \tau^e_{n,i}\}_{i \leq |C_n|}$ and their timestamps, where each caption $c_{n,i}$ describes a segment of the video, that starts at $\tau^s_{n,i}$ and ends at $\tau^e_{n,i}$.

The generated captions aim to serve for vision-language or vision-language-{other modalities (such as audio)} tasks, providing language supervision in the form of "human-written-like" captions rather than scrambled noisy ASR subtitles. That enables the potential of collecting large-scale datasets with long-term videos and their fine-grained textual descriptions for free, without human supervision.

### 3.2 VIDEO-LANGUAGE RETRIEVAL MODEL

Before we will describe our method for generating the HowToCaption dataset, we will briefly recap the training of video-language retrieval models (V-L model), as it is one of the main use cases for this dataset. Furthermore, we also use a V-L model to improve the temporal alignment in the dataset.

We base our video-language retrieval model (V-L model) on the pre-trained BLIP image-language dual encoder model (Li et al., 2022). We maintain the architecture of the text encoder $f(c) \in \mathbb{R}^d$ but, following CLIP4CLIP (Luo et al., 2022), adapt the image encoder $g(I)$ to a video encoder by averaging image embeddings obtained from uniformly sampled frames of the video: $g(V_n) = \sum_{I \in V_n} g(I) \in \mathbb{R}^d$. Dual encoder models typically learn a cross-modal embedding space (Radford et al., 2021; Luo et al., 2022) via training with the symmetric InfoNCE loss (Oord et al., 2018). The training is based on a similarity metric (often cosine distance) between embeddings $\rho_{n,i,m} = \text{sim}(f(c_{n,i}), g(V_m))$ scaled by a temperature parameter $\nu$, resulting in the following loss function:

$$L = -\frac{1}{2|B|} \sum_{(n,i) \in B} \left( \log \frac{\exp(\rho_{n,i,n}/\nu)}{\sum_{(m,j) \in B} \exp(\rho_{n,i,m}/\nu)} + \log \frac{\exp(\rho_{n,i,n}/\nu)}{\sum_{(m,j) \in B} \exp(\rho_{m,j,n}/\nu)} \right) \tag{1}$$

where $B$ is a batch of training sample indices $(n, i)$.

### 3.3 HOWTOCAPTION

To generate fine-grained captions for the instructional videos, we propose to leverage recent large language models that demonstrate great zero-shot performance in many different tasks formulated with natural language. Namely, we prompt the LLM to read the ASR subtitles of the video and create a plausible video description based on this. Since one subtitle only covers a small part of the video and lacks a global context, we propose to aggregate multiple subtitles together with their

timestamp information. Then, we task the LLM to create detailed descriptions based on the entire input and estimate timestamps for each generated sentence.

The overview of our approach is shown in Figure 1. For each video, first, we slice a given sequence of subtitles into blocks that contain long context information about the video. Then, the ASR subtitles of each block are summarised into a video caption using the LLM that we prompt with our task description. The LLM also predicts timestamps for each sentence in the video caption, which we further refine in our post-processing step based on similarities of a caption sentence to video clips in the neighboring area of predicted timestamps.

### 3.3.1 LLM Prompting

For our language prompt (shown in Figure 1), we leverage the same "main" prompt for LLM, as in the Vicuna-13B model (Chiang et al., 2023): "A chat between a curious human and..." that defines the requirement from LLM to give a helpful answer to our questions. Then, we describe our request, what data we need to process and how it should be processed: "I will give you an automatically recognized speech...". We found structuring the prompt in the way that the task description given at the beginning of the prompt and the long ASR input $S_n$ at the end is beneficial. Then, we give detailed instructions about how to process ASR subtitles. We found that instructions such as "Write only short sentences" or "Describe only one action per sentence" are beneficial, as they encourage the creation of concise captions that better match the video content. The instruction "Keep only actions that happen in the present time" is intended to filter out unrelated chats, advice, or comments from the captions; we observed that it also resulted in performance enhancements. Lastly, we request the model to predict a timestamp for each generated caption and, finally, input timestamps + ASR subtitles that need to be processed. The LLM response follows the start timestamp + caption format given in the prompt and, therefore, can be automatically parsed with a simple script into a set of captions and timestamps $C_n = \{c_{n,i}, \tau^s_{n,i}, \tau^e_{n,i}\}_{i \leq |C_n|}$, where we assign $\tau^e_{n,i} = \tau^s_{n,i} + \Delta_{sec}$, where $\Delta_{\text{sec}}$ is a constant video clip length parameter (number of seconds). Please see Section 4.3 and Appendix A.2 for a detailed evaluation of these choices.

### 3.3.2 Post-processing: Filtering & Alignment

ASR subtitles suffer from bad temporal alignment (Han et al., 2022; Miech et al., 2019). Although the LLM prompted to produce video captions can filter some noise in the ASR subtitles, some generated captions are still misaligned with the video. Therefore, inspired by the TAN method (Han et al., 2022) that automatically predicts the alignability of subtitles and matching timestamps, we further improve our obtained captions with a filtering & alignment post-processing step (Figure 1).

To this end, we utilize the video-language encoder model $(f, g)$. Given a generated caption $c_{n,i}$ and its start and end timestamps $(\tau^s_{n,i}, \tau^e_{n,i})$ that corresponds to a part of the video clip $V_n^{[\tau^s_{n,i}, \tau^e_{n,i}]}$, we use the V-L model to compute an alignment similarity score $\rho_{n,i}(\delta) = \text{sim}\left(f(c_{n,i}), g(V_n^{[\tau^s_{n,i}+\delta, \tau^e_{n,i}+\delta]})\right)$ between the caption and video clip with time offsets $\delta \in \mathbb{Z}, |\delta| \leq T$ around predicted timestamps. Then we *align* the caption to the video clip by finding the most similar clip of the video around the timestamp $\delta^*_{n,i} = \underset{\delta \in \{-T,...,T\}}{\arg\max} \rho_{n,i}(\delta)$ and *filter* out pairs if $\rho_{n,i}(\delta^*_{n,i}) < \kappa$, where $\kappa$ is a similarity score threshold.

To further improve the alignment of captions, we perform multiple rounds of filtering & alignment. In practice, we found that the improvement after two rounds is marginal. For subsequent rounds, we finetune (c.f. Section 3.2) the V-L model on the filtered & aligned video-captions pairs $\{(v_i, c_i)\}$, resulting in new alignment scores $\rho'_{n,i}(\delta)$. Since finetuning V-L models often leads to forgetting, we employ two modifications in the finetuning and second alignment processes. First, during fine-tuning, we add regularization $L_{\text{align}} = \alpha \frac{1}{2|B|} \sum_{(n,i) \in B} (\text{sim}(f(c_{n,i}), f^*(c_{n,i})) + \text{sim}(g(V_n), g^*(V_n))$ where $f^*$ and $g^*$ denote frozen text and video encoders, $\alpha$ is a regularization weight, and $(n, i) \in B$ represents the samples batch $B$. This regularization prevents the model from forgetting (Hou et al., 2019). Then, during filtering & alignment, we use the average of the similarities of the finetuned and original model. We show an impact of these changes in Appendix A.3.

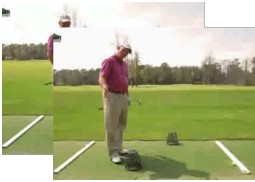 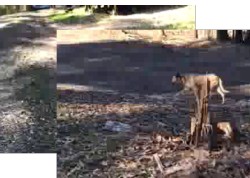 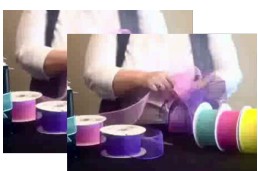 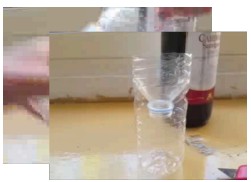

**Caption**: Matt Swanson gives a tip to use buckets to direct the path of the ball
ASR: move them around to help direct the path

**Caption**: Dog wants to hang out near dirt or other dogs with bones to acquire more bones
ASR: she might need it for later so this is stage one of hiding the bone burying the bone...

**Caption**: Making a bow with two colors
ASR: so it's not going to really show

**Caption**: Make sure the bottle stays together
ASR: but this yeah and it just stays or it won't get off it's busy here

Figure 2: **Examples video-captions pairs from our HowToCaption dataset.** ASR subtitles with only noisy supervision for the video are converted from spoken to written-language-style captions.

## 3.4 HOWTOCAPTION DATASET

We apply the proposed HowToCaption approach to 1.2M long-term instructional videos and ASR subtitles of the HowTo100M dataset and obtain the HowToCaption dataset. By prompting the Vicuna-13B model, we obtain ∼ 70M initial captions. After filtering & alignment (details in Section 4.2) we obtain 25M high-quality video-caption pairs. We show examples from our HowToCaption dataset in Figure 2. We note that generated captions follow different text styles, *e.g.*, the first and the second examples contain a long description of an object and its actions, the third describes the process, and the last one is instruction. The average length of the generated captions is 9.3 words.

## 4 EXPERIMENTAL RESULTS

To evaluate the proposed HowToCaption dataset as a video-text dataset for large-scale pre-training of vision-language models, we train our T-V model on HowToCaption and assess its zero-shot video-text retrieval performance on four widely recognized video-text benchmarks: YouCook2 (Zhou et al., 2018), MSR-VTT (Xu et al., 2016), MSVD (Chen & Dolan, 2011), and LSMDC (Rohrbach et al., 2015). While the YouCook2 dataset consists of instructional cooking videos and might be considered as an in-domain benchmark for the HowToCaption, the other datasets encompass a broader range of topics and video types, including non-instructional YouTube videos and movies. To evaluate the properties of HowToCaption dataset in comparison with other large-scale pre-training datasets, we also train our T-V model on HowTo100M (Miech et al., 2019), on HowTo100M with step labels (Lin et al., 2022), HTM-AA (Han et al., 2022), VideoCC3M (Nagrani et al., 2022), and WebVid2M (Bain et al., 2021) datasets and compare zero-shot text-video retrieval performance.

### 4.1 DATASETS AND METRICS

**Pre-training Datasets. HowTo100M** is a dataset of 1.2M instructional videos with ASR subtitles collected by querying YouTube with 23k different "how to" tasks from WikiHow articles. We consider three versions of annotations of this dataset: *Sentencified HowTo100M*, with pre-processed ASR subtitles by structuring them into full sentences by Han et al. (2022); *HowTo100M with Distant Supervision*, where ASR subtitles were linked to WikiHow (Koupaee & Wang, 2018) step descriptions via distant supervision by Lin et al. (2022); and *HTM-AA* (Han et al., 2022), an auto-aligned (AA) version of HowTo100M, where subtitle timestamps were adjusted to improve alignment to videos, discarding non-alignable subtitles. **WebVid2M** (Bain et al., 2021) is a large open-domain dataset of 2.5M of short videos scrapped from the internet with their alt-text. **VideoCC3M** (Nagrani et al., 2022) is a dataset of 10M video-text pairs collected by transferring captions from image-text CC3M dataset (Changpinyo et al., 2021) to videos with similar visual content.

**Downstream Datasets. YouCook2** (Zhou et al., 2018) is a dataset of instructional cooking videos, where each video clip is annotated with a recipe step. We used 3.5k test set for evaluation. **MSR-VTT** (Xu et al., 2016) contains 10k YouTube videos on various topics and human descriptions.

Table 1: **Ablation of LLM prompts.** We step by step construct a prompt for LLM that concisely and in detail describes the caption generation task. To emphasize our incremental adjustments, we label the sentences as x$n$ (where $n$ is an index). Each prompt consists of sentences that were already used in previous prompt versions (e.g., <x1>, <x2>) and new sentences introduced in the current prompt (e.g., x4: Write only ...). With each prompt, we obtain 2M video-text pairs from 100k HowTo100M videos that we later use for T-V model training (lower-resource setup). Downstream zero-shot text-video retrieval performance is reported.

| Prompt | YouCook2 R10↑ | YouCook2 MR↓ | MSR-VTT R10↑ | MSR-VTT MR↓ | MSVD R10↑ | MSVD MR↓ | LSMDC R10↑ | LSMDC MR↓ | **Average** R10↑ | **Average** MR↓ |
|---|---|---|---|---|---|---|---|---|---|---|
| x1: Here is an automatically recognized speech from a video: <ASR with timestamps>. x2: Write a synopsis for this video. x3: Begin each sentence with an estimated timestamp. | 37.5 | 22.5 | 71.0 | 3 | 80.5 | 2 | 37.3 | 30 | 56.6 | 14.4 |
| <x1> <x2> x4: Write only short sentences. <x3> | 39.3 | 20.5 | 71.4 | 3 | 81.0 | 2 | 36.5 | 32.5 | 57.1 | 14.5 |
| <x1> <x2> <x4> x5: Describe only one action per sentence. <x3> | 39.8 | 20 | 71.0 | 3 | 80.9 | 2 | 37.2 | 30.5 | 57.2 | 13.9 |
| <x1> <x2> <x4> <x5> x6: Keep only actions that happen in the present time. <x3> | 39.5 | 19.5 | 71.6 | 3 | 81.2 | 2 | 37.9 | 29 | 57.6 | 13.4 |
| <x1> x2': Write *a summary* for this video. <x4> <x5> <x6> <x3> | 40.4 | 19 | 71.4 | 3 | 81.4 | 2 | 37.1 | 30 | 57.6 | 13.5 |
| x1': Here is an automatically recognized speech from *a video segment that is cut from a long video*: <ASR with timestamps> x2'': Write a summary for *this video segment*. <x4> <x5> <x6> <x3> | 40.0 | 20 | 72.0 | 3 | 81.1 | 2 | 37.8 | 29 | 57.7 | 13.5 |
| *I will give you* an automatically recognized speech with timestamps from a video segment that is cut from a long video. <x2''> <x4> <x5> <x6> <x3> *Here is this automatically recognized speech:* <ASR with timestamps> | 40.6 | 19 | 72.0 | 3 | 81.6 | 2 | 37.7 | 30 | 58.0 | 13.5 |

Table 2: **Effect of a longer context.** For the "no context" option, we predict captions from individual ASR subtitles. With our "long context" option, we input multiple ASR subtitles with timestamps and the model generated captions based on longer context. This ablation is done in lower-resource setup.

| Method | YouCook2 R1↑ | YouCook2 R5↑ | YouCook2 R10↑ | YouCook2 MR↓ | MSR-VTT R1↑ | MSR-VTT R5↑ | MSR-VTT R10↑ | MSR-VTT MR↓ | MSVD R1↑ | MSVD R5↑ | MSVD R10↑ | MSVD MR↓ | LSMDC R1↑ | LSMDC R5↑ | LSMDC R10↑ | LSMDC MR↓ | Average R1↑ | Average R5↑ | Average R10↑ | Average MR↓ |
|---|---|---|---|---|---|---|---|---|---|---|---|---|---|---|---|---|---|---|---|---|
| No context: single ASR subtitle | 11.1 | 27.9 | 38.4 | 21 | 37.7 | **62.4** | **72.6** | **3** | 43.3 | 71.7 | 80.2 | **2** | 16.5 | 30.4 | **38.4** | 30 | 27.1 | 48.1 | 57.4 | 14 |
| Long context: multiple ASR+timestamps | **12.1** | **30.0** | **40.6** | **19** | **37.9** | 61.6 | 72 | **3** | **43.9** | **72.7** | **81.6** | **2** | **16.8** | **31.4** | 37.7 | 30 | **27.7** | **48.9** | **58.0** | **13.5** |

Following previous works (Bain et al., 2021; Nagrani et al., 2022), we use the 1k test set for evaluation. **MSVD** (Chen & Dolan, 2011) is a dataset of video snippets with their textual summary. The evaluation set consists of 670 videos. **LSMDC** (Rohrbach et al., 2015) is a collection of movies sliced into video clips with human-written descriptions. The test set consists of 1k video-caption pairs.

**Metrics.** To evaluate zero-shot text-video retrieval, we used standard Recall@$K$ metrics where $K \in 1, 5, 10$ (R1, R5, R10) and Median Rank (MedR).

## 4.2 IMPLEMENTATION DETAILS

As LLM, we utilize Vicuna-13B (Chiang et al., 2023), which is LLAMA (Touvron et al., 2023) model fine-tuned to follow natural language instructions. In Appendix A.1 we additionally experiment with the MiniGPT-4 model (Zhu et al., 2023) to generate captions from subtitles grounded on visual content. For our T-V model, we follow the dual encoder of the BLIP architecture. We uniformly sample 4 frames from a video clip during training and 12 frames during evaluation. For HowToCaption we use $T = 10$ seconds offset for filter&alignment and adaptive threshold $\kappa$ to leave 25M most similar pairs after filtering. Following (Chen et al., 2021) that found that 8-sec clips are optimal for training on the HowTo100M, we set $\Delta_{\text{sec}} = 8$. More details are in Appendix A.4.

## 4.3 ABLATION STUDIES

**Prompt Engineering.** Since prompting LLM with subtitles from 1.2M videos is resource extensive, we perform prompt engineering ablations in a lower-resource setup, where we use a 100k subset of HowTo100M videos ($\sim 10\%$ of all videos) to create dense captions with the LLM and use the threshold $\kappa$ to obtain the 2M most confident video-caption pairs. Here, we train the T-V model for 150k iterations and then evaluate zero-shot on downstream tasks. In Table 1, we begin with a basic prompt for an LLM, gradually refining it to generate captions more suitable for vision-language tasks. It is essential to recognize that the impact of various prompts on performance can vary across datasets, as certain prompts may yield captions better aligned with specific downstream

Table 3: **Effect of filtering&alignment.** With each post-processing variant, we obtain 25M video-text pairs that we later use for T-V model training. Downstream zero-shot text-video retrieval performance is reported.

| Caption Post-processing | YouCook2 R10↑ | YouCook2 MR↓ | MSR-VTT R10↑ | MSR-VTT MR↓ | MSVD R10↑ | MSVD MR↓ | LSMDC R10↑ | LSMDC MR↓ | Average R10↑ | Average MR↓ |
|---|---|---|---|---|---|---|---|---|---|---|
| Lower bound: original ASR as supervision | 39.3 | 20 | 61.7 | 5 | 77.1 | 2 | 31.5 | 56 | 52.4 | 20.8 |
| No post-processing | 40.2 | 18 | 65.9 | 4 | 79.8 | 2 | 34.4 | 40 | 55.1 | 16.0 |
| Filtering (using BLIP) | 42.5 | 16 | 71.2 | 3 | 81.7 | 2 | 37.4 | 30 | 58.2 | 12.8 |
| Filtering&alignment (using BLIP) | 42.4 | 17 | 71.7 | 3 | 82.2 | 2 | 38.5 | 29.5 | 58.7 | 12.9 |
| Filtering&alignment (with ours) | 44.1 | 15 | 73.3 | 3 | 82.1 | 2 | 38.6 | 29 | 59.5 | 12.3 |

Table 4: **Zero-shot text-to-video retrieval performance of model trained on different video-text datasets.** For each dataset, we train our T-V model and report downstream zero-shot text-video retrieval performance.

| Video-Text Training Data | YouCook2 R1↑ | R5↑ | R10↑ | MR↓ | MSR-VTT R1↑ | R5↑ | R10↑ | MR↓ | MSVD R1↑ | R5↑ | R10↑ | MR↓ | LSMDC R1↑ | R5↑ | R10↑ | MR↓ | Average R1↑ | R5↑ | R10↑ | MR↓ |
|---|---|---|---|---|---|---|---|---|---|---|---|---|---|---|---|---|---|---|---|---|
| - (zero-shot, with BLIP initialization) | 6.1 | 16.2 | 23.6 | 69 | 34.3 | 59.8 | 70.6 | 3 | 38.5 | 65.0 | 74.0 | 2 | 14.7 | 29.5 | 36.5 | 31 | 23.4 | 42.6 | 51.2 | 26.3 |
| HowTo100M with ASRs | 12.2 | 29.1 | 39.3 | 20 | 30.8 | 52.6 | 61.7 | 5 | 39.2 | 68.3 | 77.1 | 2 | 12.9 | 24.7 | 31.5 | 56 | 23.8 | 43.7 | 52.4 | 20.8 |
| HowTo100M with distant supervision | 8.3 | 21.5 | 30.3 | 34 | 28.6 | 54.0 | 66.3 | 5 | 38.5 | 68.6 | 79.4 | 2 | 12.1 | 24.7 | 32.4 | 42.5 | 21.9 | 42.2 | 52.1 | 20.9 |
| HTM-AA | 13.4 | 32.2 | 43.5 | 15 | 29.8 | 54.1 | 64.3 | 4 | 38.7 | 68.6 | 78.7 | 2 | 11.9 | 23.9 | 30.5 | 46 | 23.5 | 44.7 | 54.3 | 16.8 |
| HowToCaption (ours) | 13.4 | 33.1 | 44.1 | 15 | 37.6 | 62.0 | 73.3 | 3 | 44.5 | 73.3 | 82.1 | 2 | 17.3 | 31.7 | 38.6 | 29 | 28.2 | 50.0 | 59.5 | 12.3 |
| VideoCC3M | 5.3 | 15.1 | 21.7 | 84 | 33.9 | 57.9 | 67.1 | 4 | 39.6 | 66.7 | 76.8 | 2 | 14.8 | 29.4 | 35.8 | 33 | 23.4 | 42.3 | 50.4 | 30.8 |
| WebVid2M | 7.3 | 20.7 | 29.0 | 46 | 38.5 | 61.7 | 71.9 | 3 | 44.5 | 73.4 | 82.1 | 2 | 17.8 | 31.2 | 39.8 | 25 | 27.0 | 46.8 | 55.7 | 19.0 |

tasks. Notably, incorporating key phrases such as "Write only short sentences" or "Describe only one action per sentence" leads to performance improvements on 3 out of 4 datasets. Additionally, the use of the phrase "Keep only actions that happen in the present time" also resulted in performance enhancements. Furthermore, structuring the task description at the beginning and presenting the data to be processed at the end (the final modification) also boosts performance. We provide more ablations on prompt engineering in Appendix A.2. We also examine the impact of leveraging a longer context for caption prediction. In Table 2, we compare caption generation with "no context", where captions are predicted from individual ASR subtitles. In this option, timestamps of the input ASR are used as timestamps of the prediction caption. With our "long context" option, we input multiple ASR subtitles with their timestamps, and the model predicts both captions and timestamps based on longer context. We found that using a longer context is beneficial, resulting in an average improvement 0.6pp in R10, and particularly advantageous for the YouCook2 and MSVD datasets.

**Filtering & Alignment.** Further, we assess the impact of the proposed filtering&alignment procedure on the quality of captions of the acquired dataset in Table 3. We examine the performance of the T-V model when trained on differently post-processed versions of the dataset. Remarkably, we discovered that the obtained video-caption pairs, even without any post-processing, significantly outperform the original ASR-based supervision. Subsequently, by employing the filtering and alignment procedure to leave only 25M pairs based on video-caption similarities derived from BLIP pretrained weights, we achieve a notable performance enhancement of 3.6 p.p.in R10. Furthermore, filtering & alignment with our proposed fine-tuning without forgetting yields an additional 0.8pp boost in R10 performance. More ablations can be found in Appendix A.3.

## 4.4 MAIN RESULTS

**Comparision With Other Web Datasets.** In Table 4, we assess the pre-training effectiveness of our proposed HowToCaption dataset compared to other web video-language datasets. Specifically, we evaluate different textual annotations of HowTo100M videos: sentencified ASR subtitles (Han et al., 2022), task steps from distant supervision (Lin et al., 2022), and auto-aligned ASR subtitles (Han et al., 2022). Additionally, we conduct evaluations on WebVid2M (Bain et al., 2021) and VideoCC3M (Nagrani et al., 2022) datasets. Our findings indicate that the model pre-trained on our HowToCaption dataset significantly outperforms models pre-trained on other versions of HowTo100M annotations, with an average improvement of 5.2pp in R10. This improvement is most pronounced for the MSR-VTT, MSVD, and LSMDC datasets, which feature full-sentence captions. Interestingly, for the YouCook2 dataset with captions in the form of step descriptions like "cut tomato", HTM-AA already exhibits a high baseline performance, but our HowToCaption still provides a performance boost. We also observe that the VideoCC3M dataset does not enhance initial BLIP performance on any datasets except for the MSVD. We attribute it to the fact that the VideoCC3M dataset adopts captions from the CC3M dataset (Changpinyo et al., 2021) and transfers them to videos, potentially not introducing significantly new knowledge for the BLIP-initialised

Table 5: **Comparison in zero-shot text-to-video retrieval with baseline methods**: Nagrani et al. (2022), Frozen-in-Time (Bain et al., 2021), CLIP-straight (Portillo-Quintero et al., 2021), CLIP4CLIP (Luo et al., 2022), VideoCoCa (Yan et al., 2022), BLIP (Li et al., 2022). "+ fusion b." denotes a usage of a fusion bottleneck., "+ temp" denotes of usage of temporal attention. ‡CC (Changpinyo et al., 2021)+COCO (Lin et al., 2014)+VG (Krishna et al., 2017)+SBU (Ordonez et al., 2011) +LAION (Schuhmann et al., 2021).

| Method | Vision Encoder | Image-Text Data | Video-Text Data | YouCook2 R1↑ | R5↑ | R10↑ | MR↓ | MSR-VTT R1↑ | R5↑ | R10↑ | MR↓ | MSVD R1↑ | R5↑ | R10↑ | MR↓ | LSMDC R1↑ | R5↑ | R10↑ | MR↓ |
|---|---|---|---|---|---|---|---|---|---|---|---|---|---|---|---|---|---|---|---|
| Nagrani et al. (2022) | ViT-B + fusion. b. | - | VideoCC3M | - | - | - | - | 18.9 | 37.5 | 47.1 | - | - | - | - | - | - | - | - | - |
| Frozen-in-Time | ViT-B/16 + temp. | CC+COCO | WebVid-2M | - | - | - | - | 24.7 | 46.9 | 57.2 | 7 | - | - | - | - | - | - | - | - |
| CLIP-straight | ViT-B/32 | WIT | - | - | - | - | - | 31.2 | 53.7 | 64.2 | 4 | 37.0 | 64.1 | 73.8 | 2 | 11.3 | 22.7 | 29.2 | 56.5 |
| CLIP4CLIP | ViT-B/32 | WIT | HTM100M | - | - | - | - | 32.0 | 57.0 | 66.9 | 4 | 38.5 | 66.9 | 76.8 | 2 | 15.1 | 28.5 | 36.4 | 28 |
| VideoCoCa | ∼ViT-B/18 + temp. | JFT-3B | VideoCC3M | 16.5 | - | - | - | 31.2 | - | - | - | - | - | - | - | - | - | - | - |
| BLIP | ViT-B/16 | 5 datasets‡ | - | 6.1 | 16.2 | 23.6 | 69 | 34.3 | 59.8 | 70.6 | 3 | 38.5 | 65.0 | 74.0 | 2 | 14.7 | 29.5 | 36.5 | 30.5 |
| **Ours** | ViT-B/16 | 5 datasets‡ | HTM-Captions | 13.4 | 33.1 | 44.1 | 15 | 37.6 | 62 | 73.3 | 3 | 44.5 | 73.3 | 82.1 | 2 | 17.3 | 31.7 | 38.6 | 29 |

model since BLIP was pre-trained on multiple datasets including CC3M. On the other hand, Web-Vid2M demonstrated performance improvements across all datasets, but our HowToCaption dataset notably outperforms WebVid2M on YouCook2 and MSR-VTT, only underperforming on LSMDC.

**Comparison with SOTA in Zero-shot Text-Video Retrieval.** In Table 5, we also conduct a comparison with zero-shot retrieval baselines. It is important to acknowledge that comparing state-of-the-art methods can be challenging due to variations in backbone capacity, training objectives, and other factors. Nevertheless, it is worth highlighting that our approach consistently outperforms the baseline methods in zero-shot text-video retrieval across all datasets.

**Text-Video+Audio Retrieval.** It is known that instructional video datasets, e.g., HowTo100M or HD-VILA, suffer from a high correlation of audio modality to a textual description, therefore hindering building a text-video+audio retrieval system where the video is extended with audio. The usage of ASR narrations as supervisory textual description leads retrieval models to primarily perform speech recognition on the audio, hindering true language-audio connections. Therefore, training text-video+audio systems on these datasets usually requires additional regularization, such as shifting audio timestamps or assigning lower weights to the audio loss (Shvetsova et al., 2022). Our How-

Table 6: **Zero-shot text-video+audio retrieval.** MIL-NCE (Miech et al., 2020), TAN (Han et al., 2022), MMT (Gabeur et al., 2020), AVLNet (Rouditchenko et al., 2021), MCN (Chen et al., 2021), EAO (Shvetsova et al., 2022). ‡ denote text-video only retrieval models. R152+RX101 denotes ResNet-152+ResNeXt101.

| Method | Vision Enc | YouCook2 R1↑ | R5↑ | R10↑ | MR↓ | MSR-VTT R1↑ | R5↑ | R10↑ | MR↓ |
|---|---|---|---|---|---|---|---|---|---|
| MIL-NCE‡ | S3D | 15.1 | 38.0 | 51.2 | 10 | 9.9 | 24.0 | 32.4 | 29.5 |
| TAN‡ | S3D | 20.1 | 45.5 | 59.5 | 7.0 | - | - | - | - |
| MMT | Transformer | - | - | - | - | - | 14.4 | - | 66 |
| AVLNet | R152+RX101 | 19.9 | 36.1 | 44.3 | 16 | 8.3 | 19.2 | 27.4 | 47 |
| MCN | R152+RX101 | 18.1 | 35.5 | 45.2 | - | 10.5 | 25.2 | 33.8 | - |
| EAO | S3D | 24.6 | 48.3 | 60.4 | 6 | 9.3 | 22.9 | 31.2 | 35 |
| Ours | S3D | 25.5 | 51.1 | 63.6 | 5 | 13.2 | 30.3 | 41.5 | 17 |

ToCaption dataset resolves this issue by providing richer textual descriptions, allowing us to train a text-video+audio retrieval system without regularization. To evaluate this, we train a multimodal Everything-At-Once (EAO) (Shvetsova et al., 2022) model that learns to fuse any combinations of text, video, and audio modalities on our proposed HowToCaption without any additional tricks and evaluate zero-shot text-video+audio retrieval performance. Table 6 shows the proposed model significantly outperforms all baselines and over directly comparable EAO model.

# 5 CONCLUSION

Freely available web videos serve as a rich source of multimodal text-video data. Nevertheless, training on such data presents challenges, primarily due to weak supervision offered by video subtitles for text-visual learning. In this method, we address this problem by leveraging the capabilities of large-language models (LLMs). We propose a novel approach, HowToCaption, that involves prompting an LLM to create detailed video captions based on ASR subtitles. Simultaneously, we temporally align the generated captions to videos by predicting timestamps with LLM that is further followed by the filtering & alignment step, which additionally ensures synchronization with the video content. To validate the efficacy of the proposed HowToCaption method, we curate a new large-scale HowToCaption dataset, featuring high-quality human-style textual video descriptions derived from the videos and ASR subtitles of the HowTo100M dataset. Our HowToCaption dataset helps to improve performance across multiple text-video retrieval benchmarks and also separates textual subtitles from the audio modality, enhancing text-to-video-audio tasks. This work demonstrates the potential of LLMs for creating annotation-free, large-scale text-video datasets.

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

## A   APPENDIX

In the appendix, we provide additional experimental evaluations and additional implementation and dataset details. First, we conduct experiments with MiniGPT-4 (Zhu et al., 2023) to generate captions that are grounded on visual content in Appendix A.1. Then, we provide additional results of prompt engineering in Appendix A.2, perform ablation of our filtering & alignment method in Appendix A.3, and provide additional implementation details in Appendix A.4. Finally, we provide statistics and additional qualitative examples of the HowToCaption dataset in  Appendix A.5 and Appendix A.6.

### A.1   GROUNDING CAPTIONS TO VIDEO CONTENT WITH MINIGPT-4

Our generated captions with Vicuna-13B are based solely on ASR subtitles. To additionally ground the produced captions on visual content, we experiment with the recent miniGPT-4 model (Zhu et al., 2023). The MiniGPT-4 consists of the frozen Vicuna-13B model and a visual encoder with a Q-Former Li et al. (2022) that projects visual features from an image into tokens in a language model embedding space that are later treated as word tokens in the Vicuna-13B model. To ground generated captions in the visual modality, we create a grid image from 4 uniformly sampled frames from a video clip and slightly adapt the prompt to enforce the LLM to leverage the given image into generated captions (Table 7). We applied our approach to obtain visually grounded captions with the MiniGPT-4 model and obtain *HowToCaption-grounded*. For this dataset, we follow exactly the same hyperparameters that we use for HowToCaption. In Table 8, we evaluate the downstream retrieval performance of the T-V model trained on HowToCaption-grounded. The dataset shows mixed results compared to the HowToCaption; while it is beneficial for the MSR-VTT and the MSVD dataset, performance on the YouCook2 dataset drops. To facilitate further analysis, we will release both captions sets: the ASR-based only HowToCaption, produced by the Vicuna-13B, and HowToCaption-grounded, produced by the MiniGPT-4.

### A.2   ADDITIONAL RESULTS IN PROMPT ENGINEERING

In Table 9, we provide an additional evaluation of language prompts. First, we experiment with phrases such as "write a *likely* summary..." and "write a *creative* summary...". While the keyword "likely" almost does not change downstream performance, the keyword "creative" is not beneficial for 3 out of 4 datasets. We also experiment with utilizing another timestamp format in the LLM prompt. Namely, instead of using *"n"s* (such as 0s, 65s), we use *"minutes":"seconds"* format (such as 00:00, 01:05). We found that simple timestamp format *"n"s* results in higher performance.

### A.3   ABLATIONS OF FILTERING & ALIGNMENT POST-PROCESSING

In Table 10, we ablate two modifications of the fine-tuning and alignment processes for the second round of filtering & alignment. We observe that the dataset obtained after the second round of filtering & alignment without these modifications shows lower performance than the dataset obtained with the first round (using the BLIP model). We attribute this to forgetting during fine-tuning. However, we note that both proposed modifications boost performance, as well as their combination. In Table 11, we also analyze if more rounds of filtering & alignment lead to a better quality dataset. We employ 20k iterations of fine-tuning of the T-V model on the obtained dataset after each filtering & alignment round. We do not observe any performance boost with more filtering & alignment rounds.

### A.4   ADDITIONAL IMPLEMENTATION DETAILS

For our T-V model, we follow BLIP's (Li et al., 2022) dual encoder architecture with a ViT-B/16 visual encoder and a BERT$_{base}$ textual encoder, which are initialized with BLIP$_{CapFilt-L}$ pre-trained weights. Following BLIP (Li et al., 2022), we also use an extension of the loss Equation (1) with soft labels produced by a momentum encoder and a memory bank that keeps additional text and video embeddings from the previous iterations. We train the model for 300k iterations using AdamW (Loshchilov & Hutter, 2019) with a batch size of 128, a learning rate of 1e-6, and a weight decay of 0.05. We use a memory bank of 2048 and smooth labels with a parameter of 0.6. Training

Table 7: **Prompts for the Vicuna-13B and MiniGPT-4 models.** Difference is highlighted with bold.

| Vicuna-13B | MiniGPT-4 |
|---|---|
| I will give you an automatically recognized speech with timestamps from a video segment that is cut from a long video. Write a summary for this video segment. Write only short sentences. Describe only one action per sentence. Keep only actions that happen in the present time. Begin each sentence with an estimated timestamp. Here is this automatically recognized speech: \<ASR with timestamps\> | I will give you an automatically recognized speech with timestamps and an image with four frames from a video segment that is cut from a long video. Write a summary for this video segment **based on both: video frames and speech.** Write only short sentences. Describe only one action per sentence. Keep only actions that happen in the present time. Begin each sentence with an estimated timestamp. **Here is the image with four frames: \\<grid-image here\>\</Img\>.** Here is the automatically recognized speech: \<ASR with timestamps\> |

Table 8: **Comparison of HowToCaption and HowToCaption-grounded datasets obtained with Vicuna-13b and MiniGPT-4 large language models, respectively.** For each dataset, we train a T-V model and report downstream zero-shot text-video retrieval performance.

| Dataset | YouCook2 | | | | MSR-VTT | | | | MSVD | | | | LSMDC | | | |
|---|---|---|---|---|---|---|---|---|---|---|---|---|---|---|---|---|
| | R1↑ | R5↑ | R10↑ | MR↓ | R1↑ | R5↑ | R10↑ | MR↓ | R1↑ | R5↑ | R10↑ | MR↓ | R1↑ | R5↑ | R10↑ | MR↓ |
| HowToCaption (Vicuna-13B) | 13.4 | 33.1 | 44.1 | 15 | 37.6 | 62.0 | 73.3 | 3 | 44.5 | 73.3 | 82.1 | 2 | 17.3 | 31.7 | 38.6 | 29 |
| HowToCaption-grounded (MiniGPT-4) | 12.4 | 29.8 | 39.9 | 20.5 | 38.3 | 62.5 | 73.2 | 2 | 46.2 | 73.9 | 82.5 | 2 | 16.8 | 31.0 | 38.7 | 27 |

augmentation is cropping with a scale [0.5, 1]. For model fine-tuning in the filter & alignment step, we use 20k training iterations and regularization parameter $\alpha = 0.1$.

## A.5 HOWTOCAPTION DATASET STATISTICS

In this section, we present the statistics of our HowToCaption dataset. Our goal is to demonstrate the scale and diversity of the captions in the proposed dataset.

**Caption Length.** To better understand the scale of our dataset, we compute caption length statistics. We analyze captions both at the video clip level and at the video level (when combining captions from all clips belonging to the same video). We randomly sample 5000 videos from HowToCaption and use a spaCy tokenizer (Honnibal et al., 2020) to count words. The resulting histograms of caption length are shown in Figure 3 and statistics in Table 12. On a sentence level, our dataset has shorter captions on average (9.03 words) compared to the original ASR subtitles (10.97 words). Our captions also have a smaller standard deviation (4.36 vs. 7.91), indicating a more consistent length distribution. Note that the average word counts per caption slightly differ here from Section 3.4 because we use the spaCy tokenizer for this analysis.

**Language Diversity.** We also compare the language diversity of the ASR subtitles and our captions. We measure language diversity from two perspectives: 1) diversity based on the presence of distinct words or verbs and 2) diversity of word/verb n-grams across the captions, providing insights into the varied combinations of words/verbs used in our captions.

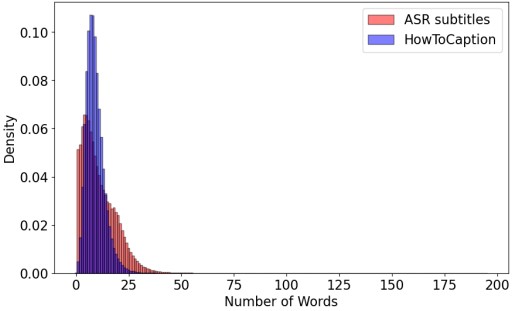

Histogram of caption lengths.

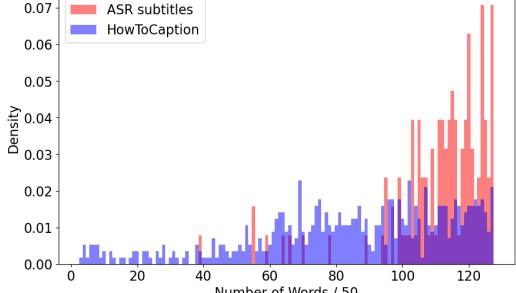

Histogram of total length of all captions for a video. The scale of the x-axis is divided by 50.

Figure 3: **Caption length statistics of our HowToCaption dataset.** We randomly sample 5000 videos to plot the distributions.

Table 9: **Additional experiments with LLM prompts.** We report modifications that we have done compared to our default prompt, which is highlighted . With each prompt, we obtain 2M video-text pairs from 100k HowTo100M videos that we later use for T-V model training (low-recourse setup). Downstream zero-shot text-video retrieval performance is reported.

| Prompt | YouCook2 | | MSR-VTT | | MSVD | | LSMDC | | **Average** | |
|---|---|---|---|---|---|---|---|---|---|---|
| | R10↑ | MR↓ | R10↑ | MR↓ | R10↑ | MR↓ | R10↑ | MR↓ | R10↑ | MR↓ |
| I will give you an automatically recognized speech with timestamps from a video segment that is cut from a long video. Write a summary for this video segment. Write only short sentences. Describe only one action per sentence. Keep only actions that happen in the present time. Begin each sentence with an estimated timestamp. Here is this automatically recognized speech: <ASR with timestamps in the format "n"s: "ASR"> **(ours)** | 40.6 | 19 | 72.0 | 3 | 81.6 | 2 | 37.7 | 30 | 58.0 | 13.5 |
| Modification: Write a summary for this video segment. ⟶ Write a **likely** summary for this video segment. | 40.8 | 18.5 | 71.4 | 3 | 81.5 | 2 | 37.7 | 30 | 57.9 | 13.4 |
| Modification: Write a summary for this video segment. ⟶ Write a **creative** summary for this video segment. | 40.0 | 19 | 71.6 | 3 | 81.2 | 2 | 37.8 | 27 | 57.7 | 12.8 |
| Modification: <ASR with timestamps in the format "n"s: "ASR"> ⟶ <ASR with timestamps in the format **"minutes":"seconds": "ASR subtitle"**> | 40.8 | 18.5 | 71.5 | 3 | 81.2 | 2 | 37.2 | 29 | 57.7 | 13.1 |

Table 10: **Ablation of our filtering & alignment method.** With each post-processing variant, we obtain 25M video-text pairs that we later use for T-V model training. Downstream zero-shot text-video retrieval performance is reported.

| Caption Post-processing | YouCook2 | | MSR-VTT | | MSVD | | LSMDC | | **Average** | |
|---|---|---|---|---|---|---|---|---|---|---|
| | R10↑ | MR↓ | R10↑ | MR↓ | R10↑ | MR↓ | R10↑ | MR↓ | R10↑ | MR↓ |
| Filtering&alignment (using the BLIP) | 42.4 | 17 | 71.7 | 3 | 82.2 | 2 | 38.5 | 29.5 | 58.7 | 12.9 |
| Filtering&alignment after second round | 42.4 | 17 | 69.4 | 3 | 81.2 | 2 | 38.1 | 33 | 57.8 | 13.8 |
| + regularization $L_{align}$ | 44.3 | 15 | 71.9 | 3 | 81.9 | 2 | 39 | 28 | 59.3 | 12 |
| + averaging similarities of the finetuned and original model | 43.7 | 15 | 72.8 | 3 | 82 | 2 | 39.6 | 27 | 59.5 | 11.8 |
| +regularization $L_{align}$ + averaging similarities of the finetuned and original model **(ours)** | 44.1 | 15 | 73.3 | 3 | 82.1 | 2 | 38.6 | 29 | 59.5 | 12.3 |

In our analysis (Table 12), we follow Goldfarb-Tarrant et al. (2020) and calculate the percentage of diverse verbs (that are not in the top 5 most frequent verbs) relative to all verbs. Following Shetty et al. (2017), we also compute the unique-to-total ratio for word unigrams, bigrams, and trigrams (e.g., the ratio between the number of unique word unigrams to the total number of word unigrams over all captions). We further use the spaCy toolkit (Honnibal et al., 2020) to extract and lemmatize verbs and calculate the unique-to-total ratio for *verb* unigrams, bigrams, and trigrams. The results in Table 12 show that the captions in the HowToCaption dataset have higher language diversity than ASR subtitles across almost all measures except on verb unigram. We observe that the longer action sequences in HowToCaption are more diverse than ASR subtitles, which demonstrates the high quality of our dataset.

## A.6  QUALITATIVE EXAMPLES

We demonstrate additional video-text examples of our HowToCaption dataset in Figure 4 and Figure 5. In Figure 5, we also showcase instances of failure cases. One such case involves a failure where the LLM was unable to generate a caption and instead copied the input ASR subtitles: "DP Move Safe lets operators get out of the classroom..." However, in this example, the ASR subtitles contain a third-person description with a subjet+verb+object sentence structure that justifies the coping input description without modification. Other failure cases include video-caption pairs, where the caption corresponds to the video only partially, e.g., "Cover it with lid" action is not visible on the video while "until the seviayan is cooked" is visible.

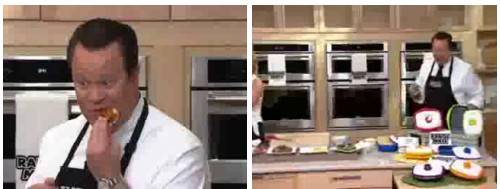

**Caption**: Video segment starts with a shot of David's face, which is described as funny
ASR: and the bottom is actually has holes in it because it gets so incredibly hot so you cannot submerge it in water so we ask you to just rinse it out real quick look at david's face he is so funny

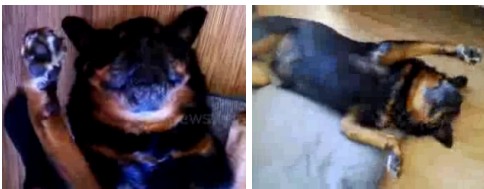

**Caption**: Brutus is encouraged to swallow his medication
ASR: if i put it in a piece of food he'll chew it up and spit it out he knows oh baby i've never seen him do this though

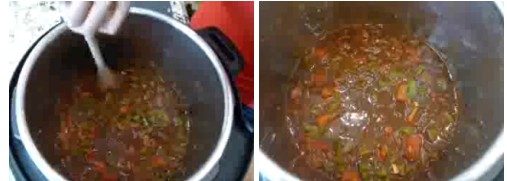

**Caption**: Adds two cans of red kidney beans to the chili
ASR: you could also use a vegetable broth all right so we're mixing this well

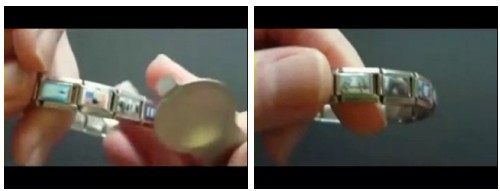

**Caption**: She explains the charm tool's little piece of metal acts as a spacer to hold the charms open
ASR: has this little piece of patootie metal right here that acts as a spacer to hold the charms open so that gap is visible in the back so it's easier to slip the charms on and off

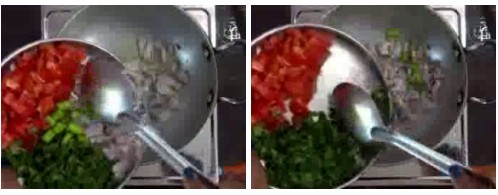

**Caption**: Adding chopped onions and green chillies to the pan
ASR:once the oil is hot enough we will add our onions and green chillies we need to cook the onions for some time maybe like 2 to 3 minutes until you start noticing that the colors of the onion have changed

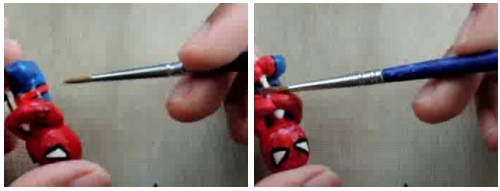

**Caption**: Paints the top part
ASR: i also notice how the blue continues onto the front of him just like right there so be careful with that next you take the white color and you would paint the webbing that he's hanging from here and also his eyes

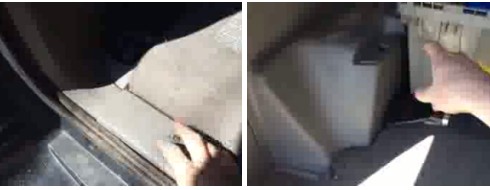

**Caption**: Shows where wire runs along inside of vehicle
ASR: then ran alongside the gasket right here and runs down here and then this we took off and then ran the wiring in through here put this back down

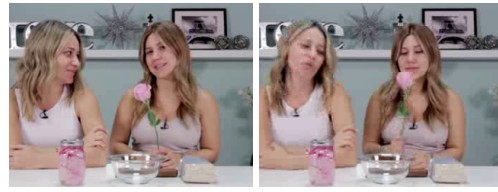

**Caption**: You are making a rose petal exfoliating face scrub
ASR: you guys one of our favorite diys ever had to do with rose petals so we thought let's make another one

Figure 4: **Examples video-captions pairs from our HowToCaption dataset.** Since ASR subtitles' timestamps do not always correspond to the timestamps of video clips from the HowToCaption dataset, we show ASR subtitles that intersect with video clip boundaries.

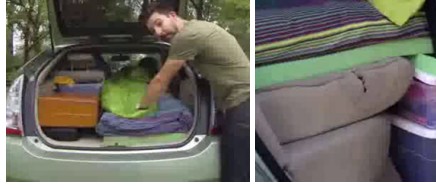 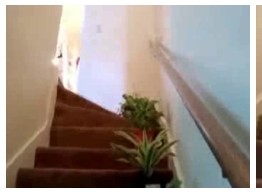 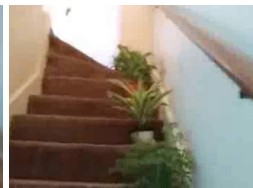

**Caption**: Soft bed in the car

**ASR**: but honestly when people see a lone prius in the parking lot no one thinks hey i wonder if someone's sleeping in there because come on it's a prius what fits in my car i have a soft bed i have a closet blackout curtains a desk kitchen table and chair a pantry a bike a laundry basket travel kit for emergencies

**Caption**: Walk upstairs to show light in the ceiling

**ASR**: i placed them over here because it's a little bit lighter on this side of the stair case then the other side there's only one light in the ceiling here so i'm gonna walk upstairs and i'm gonna let you see it from the top of the stairs one more time

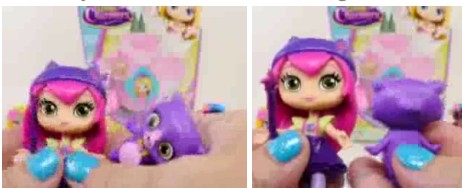 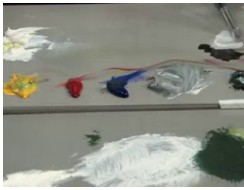 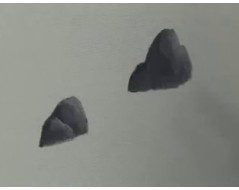

**Caption**: Asks viewers to choose favorite pet

**ASR**: i think i like them all for different reasons so that's hard so they can all three be my favorite can't they and look at her little friend

**Caption**: The speaker adds white paint to the brush to keep the color bright

**ASR**: so i lay it on with the flat of the brush which deposits it a little heavier it holds up a little better and notice i keep adding white as i'm picking up brown on this brush from the rock i have to keep adding white into my color to keep it bright enough and so you can put different angles in here

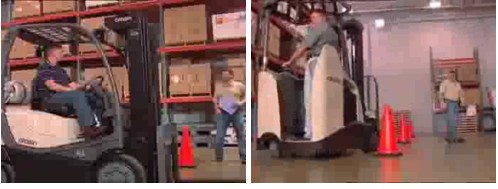 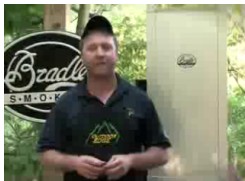 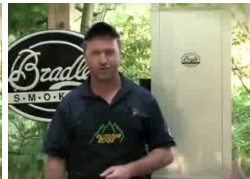

**Caption**: DP Move Safe lets operators get out of the classroom and out of their truck faster where they learn how to perform every task and do it safely (failure)

**ASR**: dp move safe lets operators get out of the classroom and out of their truck faster where they learn how to perform every task and do it safely

**Caption**: Outdoor Edge has instructional gated processing DVDs available on their website (failure)

**ASR**: this is one of the big issues with large diameter sausage products remember processing your own wild game animal can be fun easy and very rewarding if you have the tools and the knowledge to do the job you're watching outdoor edges

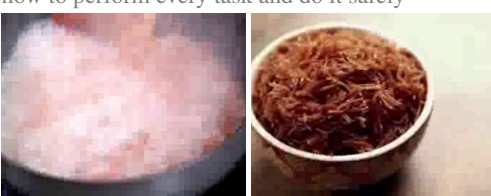 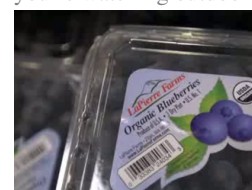 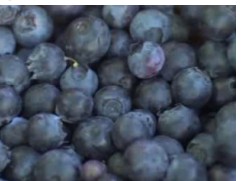

**Caption**: Cover it with a lid for 15 minutes until the seviayan is cooked (failure)

**ASR**: today, we will prepare a sweet recipe.. ..called 'sevaiyan' (vermicelli) so come on, let's see how to make sweet sevaiyan.

**Caption**: The group harvests fresh berries from the farm (failure)

**ASR**: the kids and i are here at a local blue red patches to manage to harvest some fresh berries at pcc are fresh and frozen organic blueberries help from here farms in its zilla washington their berries are so sweet we buy the entire crop

Figure 5: **Examples video-captions pairs from our HowToCaption dataset.** Failure cases are marked as (failure) . Since ASR subtitles' timestamps do not always correspond to the timestamps of video clips from the HowToCaption dataset, we show ASR subtitles that intersect with video clip boundaries.

Table 11: **Ablation of the filtering & alignment method with more rounds of filtering & alignment with fine-tuning of the T-V model after each round.** With each post-processing variant, we obtain 25M video-text pairs that we later use for T-V model training. Downstream zero-shot text-video retrieval performance is reported.

| Caption Post-processing | YouCook2 | | MSR-VTT | | MSVD | | LSMDC | | Average | |
|---|---|---|---|---|---|---|---|---|---|---|
| | R10↑ | MR↓ | R10↑ | MR↓ | R10↑ | MR↓ | R10↑ | MR↓ | R10↑ | MR↓ |
| Filtering&alignment (using the BLIP) = 1 round | 42.4 | 17 | 71.7 | 3 | 82.2 | 2 | 38.5 | 29.5 | 58.7 | 12.9 |
| Filtering&alignment after 2'nd round (**ours**) | 44.1 | 15 | 73.3 | 3 | 82.1 | 2 | 38.6 | 29 | 59.5 | 12.3 |
| Filtering&alignment after 4'th round | 44.5 | 15 | 72.2 | 3 | 81.8 | 2 | 38.6 | 29 | 59.3 | 12.3 |

Table 12: **Language statistics.** $|V|$ is the vocabulary size. #word/caption is the number of words per caption. #word/video is the number of words per all captions in a video. %diverse verb is the percentage of diverse verbs. All numbers are obtained from 5000 randomly sampled videos.

| Dataset | | Standard Statistics | | Diversity | $n$-grams Diversity | | | Verb $n$-grams Diversity | | |
|---|---|---|---|---|---|---|---|---|---|---|
| | $|V|$ | #word/caption | #word/video | %diverse verb ↑ | 1-gram↑ | 2-gram↑ | 3-gram↑ | 1-gram↑ | 2-gram↑ | 3-gram↑ |
| ASR subtitles | 45905 | 10.96 | 909.34 | 77.09 | 1.01 | 17.57 | 50.24 | **1.13** | 19.88 | 61.64 |
| HowToCaption | 36204 | 9.03 | 581.27 | **82.55** | **1.25** | **21.36** | **53.95** | 1.01 | **25.9** | **76.65** |

