# OpenReview forum: "HowToCaption: Prompting LLMs to Transform Video Annotations at Scale"
_ICLR.cc/2024/Conference — ICLR 2024 Conference Withdrawn Submission_

### Official Review · Reviewer_hb9c · 2023-10-24

**Soundness:** 2 fair
**Presentation:** 1 poor
**Contribution:** 1 poor
**Rating:** 5
**Confidence:** 5

**Summary:**

Freely available web videos provide a rich source of multimodal text-video data. However, training on such data presents challenges, primarily due to the limited guidance offered by video subtitles for text-visual learning.
In this paper, it tackle this issue by harnessing the capabilities of large-language models (LLMs). The paper introduces a novel method called HowToCaption, which entails instructing an LLM to generate detailed video captions based on automatic speech recognition (ASR) subtitles.
To assess the effectiveness of the proposed HowToCaption method, the authors have curated a comprehensive HowToCaption dataset.This research showcases the potential of large-language models in creating annotation-free, large-scale text-video datasets.

**Strengths:**

- They propose  a novel dataset HowToCaption with high-quality human-style textual descriptions.

- The paper is easy to understand.

- This paper proposes a HowToCaption method to efficiently leverages recent advances in LLMs and generates high-quality video captions at scale without any human supervision.

**Weaknesses:**

- Novelty is insufficient.  This paper propose a approach, HowToCaption, prompting an LLM to create detailed video captions based on ASR subtitles.  However, similar methods have been used widely in dataset industry, including Tencent, OpenAI and so on, which could improve the worker's efficiency. Additionally, the process of method is so engineering and not suitable for ICLR.

- Experiment  is insufficient.  Firstly, in ablation study, the authors need to evaluate different speech recognition methods. Also they don't compare the influences of  captions length.  Secondly, Other downstream tasks such as VQA and Video Caption should be performed to prove the efficiency of dataset. the paper only test it on text-video retrieval which is far from enough.

- Analysis is not enough. The setting of MSVD is not describled clearly.  the difference between HowTo100M and other large scale caption dataset such as WebVid10M  is also not describled. Also, this paper lacks some quantitative analysis  of HowToCaption, such as the distribution of caption length and the . Some settings are not be evaluate such as WebVid2M + CC3M, which is used widely in pre-trained field.

- Writing. In the process of reading, the same name of dataset and method sometimes could make reader confused. Apart from that, there are some grammar errors in paper, such as in third paragraph in 3.3, "first" is wrong.

**Questions:**

the same as weakness

---

> ### Author Response · Authors · 2023-11-14
> **Response to Reviewer hb9c  — 1/2**
>
> We thank the reviewer for providing constructive feedback on our work. In order to adequately address the reviewer's concerns, we would be grateful if the reviewer could elaborate and provide additional details regarding the potential weaknesses of our submission.
>
>
> > Novelty is insufficient. This paper propose a approach, HowToCaption, prompting an LLM to create detailed video captions based on ASR subtitles. However, similar methods have been used widely in dataset industry, including Tencent, OpenAI and so on ...
>
> If the reviewer could provide relevant references or suggestions regarding other methods to compare to that we might have missed, we are happy to provide additional comparisons to and discussions of those methods.
>
> Moreover, we would like to highlight that we pioneer a novel method involving data collection with LLMs for text-video learning, offering significant advancements in the field. This stands in contrast to recent works in prompt engineering [1,2], which focus on prompting LLMs to address pre-existing tasks. We also release a new large-scale dataset, HowToCaption, which contains high-quality video descriptions for instructional videos. Please also note that this is why we chose "Dataset and Benchmark" as the primary subject area, which is in line with the ICRL CfP.
>
> [1] Kojima, Takeshi, et al. "Large language models are zero-shot reasoners." NeurIPS, 2022.
>
> [2] Zhou, Yongchao, et al. "Large language models are human-level prompt engineers." ICLR 2023
>
> > Firstly, in ablation study, the authors need to evaluate different speech recognition methods.
>
> We thank the reviewer for the suggestion. In our work, we utilized subtitles and timestamps, officially released with the HowTo100M dataset[3], that were further postprocess into full sentences by [4] (Sentencified HowTo100M). In our preliminary study, we also experimented with the recently released WhisperX model[5] to obtain better quality subtitles and with the original officially released HowTo100M[3] subtitles. We found that all subtitles show similar performance. We report these preliminary results (note that this study is done only on 10k HowTo100M videos):
>
> |                           |   R1 |   R5 |  R10 |   MR |
> |---------------------------|-----:|-----:|-----:|-----:|
> | Original ASR subtitles [3] | 26.6 | 46.5 | 55.1 |   17 |
> | Sentencified subtitles [4] | 26.6 | 46.5 | 55.3 | 16.8 |
> | WhisperX[5] subtitles      | 26.2 | 46.4 | 55.3 | 16.6 |
>
> We observe that the quality of different ASR methods does not significantly alter the quality of the acquired dataset. We assume that the noise introduced in ASR-video alignment is inherent to the nature of speech, often not aligning precisely with the content of the video. Moreover, filtering and alignment postprocessing that filter noisy captions might further mitigate the difference. We will include it in the paper.
>
> [3] Miech, Antoine, et al. "Howto100m: Learning a text-video embedding by watching hundred million narrated video clips." ICCV, 2019.
>
> [4] Han, Tengda, Weidi Xie, and Andrew Zisserman. "Temporal alignment networks for long-term video." CVPR, 2022.
>
> [5] Bain, Max, et al. "WhisperX: Time-accurate speech transcription of long-form audio." INTERSPEECH, 2023.

---

> ### Author Response · Authors · 2023-11-14
> **Response to Reviewer hb9c  — 2/2**
>
> >  Also they don't compare the influences of captions length.
>
> Following the suggestion, we assess the length of the generated captions for different prompts from Table 1. We note that we explicitly prompt LLM to generate "short sentences" in sentence <x4>: "Write only short sentences," which contributes to improved performance. We provide here results for the subset of prompts. We will add the full table in the paper.
> | prompt               | average caption length |  R10  |  MR  |
> |----------------------|:--------------:|:-----:|:----:|
> | <x1><x2><x3>         |      15.7      | 56.6  | 14.4 |
> | <x1><x2>**<x4>**<x3>     |           11.2 | 57.1  | 14.5 |
> | <x1><x2>**<x4>**<x5><x3> | 10.1           | 57.2  | 13.9 |
> | Final prompt (the last line of Table 1) | 9.3            | 58.0  | 13.5 |
>
> > Other downstream tasks such as VQA and Video Caption should be performed to prove the efficiency of dataset. the paper only test it on text-video retrieval which is far from enough.
>
> We value the suggestion and are initiating further evaluations. While we aim to present results within the discussion period, a thorough analysis may require additional time. In the meanwhile, we provide preliminary zero-shot video clip captioning results on the YouCook2 dataset after only 1 epoch of training on 10% of the HowToCaption dataset, with the captioning model initialized from BLIP (note, we do not do any finetuning on YouCook2):
>
> |      | BLEU@4 | METEOR | ROUGE | CIDEr |
> |------|:------:|:------:|:-----:|:-----:|
> | BLIP |   0.7  |   4.4  |  9.8  |  12.6 |
> | Ours |    2.7 |    8.3 |  21.8 |  28.5 |
>
> >  The setting of MSVD is not describled clearly.
>
> For evaluation on MSVD, we follow the standard split and use 670 videos for testing with 40 captions corresponding to each video. We follow standard practice [6] and count each caption-video pair towards the metrics. We will clarify it in the paper.
>
> [6] Bain, Max, et al. "Frozen in time: A joint video and image encoder for end-to-end retrieval." ICCV, 2021.
>
> > the difference between HowTo100M and other large scale caption dataset such as WebVid10M is also not describled.
>
> We discuss the difference between large-scale datasets in Section 2.2. WebVid10M is an extension of the WebVid2M dataset, and similarly to WebVid2M, it contains only short videos that mostly don't have audio. We will add it.
>
> > Also, this paper lacks some quantitative analysis of HowToCaption, such as the distribution of caption length and the .
>
> Thanks for the suggestion! We have added more analysis of the captions of the HowToCaptions dataset in the Appendix, Section A.5 (see updated pdf). We included an analysis of caption length and language diversity.
>
> > Some settings are not be evaluate such as WebVid2M + CC3M, which is used widely in pre-trained field.
>
> We direct attention to Table 4, where we employ WebVid2M to train the dual-encoder architecture with BLIP initialization. It is important to note that BLIP initialization already encompasses CC3M. Hence, we believe this setup aligns with the reviewer's suggestion.
>
> > Writing. [...]  there are some grammar errors in paper, such as in third paragraph in 3.3, "first" is wrong.
>
> We thank the reviewer for the valuable feedback. We will proofread the paper and update it in the following revision.
>
> Please let us know if you have any other questions or concerns regarding our paper or response.

---

### Official Review · Reviewer_qBgz · 2023-10-30

**Soundness:** 3 good
**Presentation:** 2 fair
**Contribution:** 3 good
**Rating:** 5
**Confidence:** 3

**Summary:**

This work proposes a framework to improve and collect text descriptions for videos by leveraging the powerful LLMs introduced recently. It designs a prompting mechanism that asks the LLM to rephrase subtitles extracted by automatic speech recognition (ASR) from the audio. To ensure good visual-text alignment, the framework (1) prompts the LLM to output timestamps for the generated sentences and (2) utilizes a vision-language model to filter and realign sentences that are not well aligned. The ablation studies confirm the effectiveness of the prompting mechanism. Furthermore, the experiments highlight (1) the importance of filtering and realignment, and (2) the vision-language model trained on the newly collected dataset outperforms baselines in various benchmarks on the text-to-video(+audio) retrieval task.

**Strengths:**

+) Combining LLM assures the production of high-quality descriptive sentences for videos. This is an intriguing and novel approach. Moreover, prompting the LLM to assist in text-video alignment is compelling. One could speculate that the commonsense knowledge or reasoning ability inherent in LLMs could greatly enhance alignment results. The qualitative samples depicted in Figure 2 appear impressive. The authors intend to make the dataset and code publicly available.

+) The ablation study in Table 1 clearly demonstrates the design choice behind the proposed prompting mechanism. Yet, it is somewhat surprising that, as per Table 2, long context information only yields marginal improvements in downstream tasks. The analysis of the effects of filtering and alignment also highlights the necessity for robust text-clip alignment annotations.

+) The zero-shot text-to-video retrieval results depict improvements from training BLIP on the newly assembled dataset. Although the progression seems marginal compared to WebVid2M data, Table 5 showcases that BLIP training on the proposed dataset achieves state-of-the-art performance in the text-to-video retrieval task.

+) The zero-shot text-video+audio retrieval outcomes highlight the strength of the proposed dataset, as models trained on it seemingly perform better.

**Weaknesses:**

-) A primary concern is that the authors solely validate the utility of the proposed dataset for the text-video(+audio) retrieval task. Presumably, the newly collected data could be an great resource for training foundational video-text representations, video captioning models, or video question-answering systems. It's somewhat disappointing to only see results related to retrieval tasks, especially considering BLIP's capability in visual captioning and visual question-answering.

-) While the authors show that filtering/alignment augments the final retrieval performance, a direct quantification of the assembled dataset would be valuable. For instance, what is the alignment accuracy when implementing the proposed filter/alignment technique? This would help subsequent users understand the noise level they might encounter when using the dataset.

-) Merely a few qualitative examples are presented in Figures 1 and 2, with no instances of failure cases. It would be beneficial if the authors included a broader range of generated sentences in the appendix, encompassing both successful and subpar examples. Additionally, given the notorious propensity of LLMs to fabricate or mislead, this work lacks both qualitative examples and quantitative analysis to assess such issues within the proposed dataset.

-) The work lacks qualitative results for the text-video(+audio) retrieval tasks. Specifically, for the text-video+audio task, insight into whether the proposed rephrasing method can effectively circumvent issues related to overly relying on ASR-generated information would be helpful.

**Questions:**

o) Can the authors present more qualitative examples from the newly introduced dataset, including both exemplary and flawed examples?

o) Could the authors share some qualitative results for the text-video(+audio) retrieval tasks?

o) Would the authors clarify why evaluations were restricted to retrieval tasks using BLIP? Is it feasible to assess the model's performance in visual captioning or question-answering scenarios?

o) The font size in Figure 1 appears too small. Could the authors enlarge it for easier readability?

o) Considering the generality of the proposed framework, which should be applicable to any video-text dataset, have the authors considered employing these techniques across all publicly available benchmarks to train a large-scale video-language model?

---

> ### Author Response · Authors · 2023-11-14
> **Response to Reviewer qBgz**
>
> We are thankful to the reviewer for recognizing the novelty and strengths of our paper. The feedback provided is highly valued. In the following, we will try to address the remaining concerns:
>
> **Questions:**
>
> > Can the authors present more qualitative examples from the newly introduced dataset, including both exemplary and flawed examples?
>
> Thanks for the valuable suggestion! We have updated the Appendix (Section A.6) with additional qualitative examples of the HowToCaption dataset, including failure cases (see updated pdf). We aim to update the paper even with more examples in the next days.
>
> > Could the authors share some qualitative results for the text-video(+audio) retrieval tasks?
>
> We highly appreciate the suggestion. We will update the Appendix with qualitative retrieval results as well.
>
> > Would the authors clarify why evaluations were restricted to retrieval tasks using BLIP? Is it feasible to assess the model's performance in visual captioning or question-answering scenarios?
>
> We value the suggestion and are initiating further evaluations. While we aim to present results within the discussion period, a thorough analysis may require additional time. In the meanwhile, we provide preliminary zero-shot video clip captioning results on the YouCook2 dataset after only 1 epoch of training on 10% of the HowToCaption dataset, with the captioning model initialized from BLIP (note, we do not do any finetuning on YouCook2):
>
> |      | BLEU@4 | METEOR | ROUGE | CIDEr |
> |------|:------:|:------:|:-----:|:-----:|
> | BLIP |   0.7  |   4.4  |  9.8  |  12.6 |
> | Ours |    2.7 |    8.3 |  21.8 |  28.5 |
>
> For the main evaluation, we chose the retrieval setting, as it does not depend on downstream training or fine-tuning, which evaluates the influence of our dataset directly. Other downstream tasks, such as captioning, often require fine-tuning on the downstream dataset and, therefore, evaluate the pre-training dataset only indirectly, as the final performance depends on many dataset-specific hyper-parameters of training the downstream task.
>
> > The font size in Figure 1 appears too small.
>
> Thank you for pointing out the issue! We will enlarge the font in Figure 1 to make it more readable.
>
> > Considering the generality of the proposed framework, which should be applicable to any video-text dataset, have the authors considered employing these techniques across all publicly available benchmarks to train a large-scale video-language model?
>
> We appreciate your suggestion to apply our techniques across all publicly available benchmarks. We fully agree that the HowToCaption method is generalizable to other datasets. Applying it across all publicly available instructional video datasets to transform ASR into captions could significantly contribute to training foundational models at a large scale. We will release the code and dataset for our work, anticipating that it will inspire and accelerate research in this field.
>
>
> Please let us know if you have any other questions or concerns regarding our paper or response.

---

### Official Review · Reviewer_d1GH · 2023-10-31

**Soundness:** 3 good
**Presentation:** 3 good
**Contribution:** 2 fair
**Rating:** 5
**Confidence:** 3

**Summary:**

This paper presents a novel way to construct large-scale video datasets, i.e., prompting an LLM to create both natural and rich video descriptions (based on ASR narrations). In this way, this paper contributes a large-scale video dataset, HowToCaption.

**Strengths:**

1. The paper is well-written and easy to follow. Different sections are well organized to present the proposed method and the experimental results.

2. Existing large-scale video datasets typically don't include detailed annotations (in the form of dense captioning with both timestamps and captions), since the time and labor costs of annotating temporal segments would be rather expensive. This paper presents a possible solution to this issue by automating the annotation process with pre-trained LLMs.

**Weaknesses:**

1. This work is more of a prompt engineering than a research paper, since the core components, i.e., the pre-trained Large Language Model and video-language encoder are both borrowed from existing literature. In essence, the technical contribution is a little bit weak.

2. Considering the generated captions include fine-grained timestamp annotations, it would be better to evaluate the proposed method on temporal localization tasks like moment retrieval, instead of text-to-video retrieval only that doesn't require fine-grained modeling.

**Questions:**

It would be better to evaluate the proposed dataset on more challenging tasks, e.g., dense video captioning or moment retrieval.

---

> ### Author Response · Authors · 2023-11-14
> **Response to Reviewer d1GH**
>
> Thanks for highlighting the strengths of the proposed paper and for providing valuable feedback! In the following, we would like to address the remaining concerns.
>
>
> > This work is more of a prompt engineering than a research paper
>
> We would like to highlight that our main contribution is twofold: we release a new large-scale dataset, HowToCaption, which contains high-quality video descriptions for instructional videos, and we design a human-free framework to transform noisy ASR subtitles into high-quality captions for instructional videos. Essentially, enabling data collection with LLMs for text-video learning offers significant advancements in the field. This stands in contrast to recent works in prompt engineering [1,2], which focus on prompting LLMs to address pre-existing tasks. We hope that the impact of this work might, therefore, be not only a better annotation for one specific dataset but also pave the way towards a general better practice in crawling and processing large amounts of video. Please also note that this is why we chose "Dataset and Benchmark" as the primary subject area, which is in line with the ICRL CfP.
>
> [1] Kojima, Takeshi, et al. "Large language models are zero-shot reasoners." NeurIPS, 2022.
>
> [2] Zhou, Yongchao, et al. "Large language models are human-level prompt engineers." ICLR 2023
>
>
> > It would be better to evaluate the proposed dataset on more challenging tasks, e.g., dense video captioning or moment retrieval
>
> For the main evaluation, we chose the retrieval setting, as it does not depend on downstream training or fine-tuning, which evaluates the influence of our dataset directly.
> Other downstream tasks, such as captioning, evaluate the pre-training dataset only indirectly, as the final performance depends on many dataset-specific hyper-parameters of training the downstream task.
> Nonetheless, we value the suggestion and are initiating further evaluations. While we strive to present results within the discussion period timeframe, these tasks are very compute-intensive and might not finish in time. Moreover, we already provide initial results on zero-shot video clip captioning in the Response to Reviewer qBgz.
>
> Please let us know if you have any other questions or concerns regarding our paper or response.

---

### Official Review · Reviewer_MMYt · 2023-11-03

**Soundness:** 2 fair
**Presentation:** 3 good
**Contribution:** 2 fair
**Rating:** 5
**Confidence:** 4

**Summary:**

This paper proposes a new dataset HowToCaption by prompting LLMs to modify the existing ASR subtitles in HowTo100M dataset in a human readable form. The newly created dataset is then pretrained on a model and compared against existing datasets and models on zero-shot video retrieval and text-video + audio retrieval.

**Strengths:**

**Clarity:**
- The paper is well written and easy to follow.

**Significance:**
- This paper proposes a new method to reduce the mis-alignment and noise in video-text datasets without the need for human supervision. It can provide a framework for future works to create large datasets without human supervision.
- Results on text-video + audio are promising. This is generally an ignored area of research due to lack of quality datasets. Training on the proposed dataset show its effectiveness.

**Weaknesses:**

**Unclear advantages of using this dataset on video retrieval task:**
- In Table-5 the authors present a comparison with SOTA models. However, a lot of models are missing in the Table. For example LAVENDER shows 37.8 points and 46.3 points on MSRVTT and MSVD datasets respectively which is more than HowToCaption while trained much smaller data (5.5M). This begs the question why does the community need to use the proposed dataset of 25M as opposed to Vid2.5M + CC3M.

**Terminology usage:**
- In section 3.1, the authors mention that the aim is to "generate" captions. However, the term "generate" might be mis-leading as the LLM merely "rephrases" semantically the ASR subtitle within a time-step in a more human-alike sentence and doesn't add any new details.
- In section 3.3, the authors hint at "fine-grained" captions, I am not fully convinced that LLMs always produces better fine-grained captions. Sometimes, they add new unrelated details to the ASR subtitles. Since no quantifiable measure is provided in the paper, I would suggest the authors avoid this terminology.

**Questions:**

1. Are the ground truth time-stamps of ASR subtitles provided in the HowTo100M dataset? If not how are they determined?

2. In Section 4 the authors mention that dual-encoder of the BLIP is used as "T-V" model. Is it initialized with pre-trained weights of BLIP? How is it different from frozen-in-time architecture?

3. In Table-3, the authors present metrics of R-10 and MR for comparison which is rather unusual in video retrieval. Is there any specific reason for this? Is it possible to provide R-1 and R-5 scores?

---

> ### Author Response · Authors · 2023-11-14
> **Response to Reviewer MMYt  —  1/2**
>
> Thanks for taking the time to review our paper, highlighting the clarity and significance of our paper, and providing valuable feedback! We would like to address the raised points in the following:
>
> **Weaknesses**:
>
> >Unclear advantages of using this dataset on video retrieval task.
>
> Thanks for asking. LAVENDER uses a multi-modal encoder (i.e., that cross-attends video and text tokens) and only reports results after fine-tuning on each dataset.
> This differs from the evaluation in Table 5 in 2 ways: first, we focus on the comparison of dual-encoder SOTA methods (i.e., methods that do not cross-attend video and text tokens), second we consider zero-shot text-to-video performance rather than fine-tuned performance (performance of the models fine-tuned on each downstream task). We decided for this setup to evaluate the properties of the proposed dataset as a pretraining dataset for foundational models. To allow for an indirect comparison, we compare the fine-tuned performance reported by the LAVENDER method for MSR-VTT and MSVD with our zero-shot performance. It shows that pretraining with the proposed annotations provides a comparable performance even without cross-modal attention or any supervised fine-tuning.
>
> | | pre-training | fine-tuning | MSR-VTT (R1) | MSVD (R1) |
> | -------- | -------- | -------- | -------- | -------- |
> | LAVENDER | WebVid2.5+CC3M | **yes** | 37.8 | 46.3 |
> | Ours     | HowToCaption | **no** | 37.6 | 44.5 |
>
> We, therefore, assume that using our HowToCaption dataset will boost LAVENDER performance as well.
>
> We want to highlight that we intentionally do not report methods that rely on a multi-modal encoder and are trained with image-text-matching loss rather than a contrastive loss. Such methods show an improved performance compared to dual-encoder models but require significantly higher computation costs to perform retrieval. They require a full forward pass for each (video, text) pair to output the similarity of this input pair ($N^2$ forward passes to compute the similarity between $N$ text and $N$ videos), while dual-encoder methods calculate cosine similarity in embedding space ($2N$ forward passes only). For a fair comparison (and as we want to test the data rather than the method), we therefore consider dual-encoder methods.
>
> >the term "generate" might be misleading as the LLM merely "rephrases"
>
> While we task the LLM to _rephrase_ the caption, we are using the term _generate_ here as an LLM is fundamentally a generative model with no inherent guarantee that its output is aligned with the task. We will make this more clear in the paper.
>
> > In section 3.3, the authors hint at "fine-grained" captions, I am not fully convinced that LLMs always produces better fine-grained captions:
>
> We appreciate the reviewer's feedback. We agree that the term "fine-grained" might be misleading. We will revise the paper to avoid this terminology.

---

> > ### Author Response · Authors · 2023-11-14
> > **Response to Reviewer MMYt — 2/2**
> >
> > **Questions**:
> >
> > > Are the ground truth time-stamps of ASR subtitles provided in the HowTo100M dataset? If not how are they determined?
> >
> > We utilize the "Sentencified HowTo100M" version of subtitles and timestamps, released by [1]. These are subtitles that were pre-processed by structuring original subtitles officially released by the authors of HowTo100M (that might start or end in the middle of the sentence) into full sentences (by merging and separating some subtitles) and adapting timestamps accordingly. We will clarify it in the paper.
> >
> >
> > [1] Han, Tengda, Weidi Xie, and Andrew Zisserman. "Temporal alignment networks for long-term video." Proceedings of the IEEE/CVF Conference on Computer Vision and Pattern Recognition. 2022.
> >
> > > Is it initialized with pre-trained weights of BLIP? How is it different from frozen-in-time architecture?
> >
> > We initialize our T-V retrieval model from BLIP pre-trained weights and adapt the image encoder to the video encoder by averaging frames' embeddings (see Section 3.2 for a detailed explanation). It differs from the "frozen-in-time" architecture in that it does not have additional layers of temporal attention and a shared "cls" token. But extending this architecture with temporal attention in a "frozen-in-time"-manner might further boost performance.
> >
> > > In Table-3, the authors present metrics of R-10 and MR for comparison which is rather unusual in video retrieval. The authors present metrics of R-10 and MR for comparison which is rather unusual in video retrieval Is there any specific reason for this? Is it possible to provide R-1 and R-5 scores?
> >
> >
> > For some ablation studies due to space constraints, we report only R10 and MedianRank. We selected these metrics over to R1 and R5 since they are more robust for retrieval performance evaluation. The downstream datasets do not always ensure 1-to-1 text-video matching in a test set. For example, a text query "heat some oil in a pan" occurs 6 times in the YouCook2 validation set and therefore corresponds to 6 correct videos, therefore, even in the best prediction,  at least 5 out of 6 "heat some oil in a pan" text queries will get incorrect results. Therefore, R1 and R5 might be noisy. But we will add a full table to the supplement. Here, we provide all metrics for Table 3 (average over all 4 datasets).
> >
> >
> > |                                          |   R1 |   R5 |  R10 |   MR |
> > |------------------------------------------|-----:|-----:|-----:|-----:|
> > | Lower bound: original ASR as supervision | 23.8 | 43.7 | 52.4 | 20.8 |
> > | No post-processing                       | 24.5 | 45.4 | 55.1 | 16.0 |
> > | Filtering (using BLIP)                   | 27.9 | 49.0 | 58.2 | 12.8 |
> > | Filtering&alignment (using BLIP)         | 28.1 | 49.4 | 58.7 | 12.9 |
> > | Filtering&alignment (with ours)          | 28.2 | 50.0 | 59.5 | 12.3 |
> >
> >
> > Please let us know if you have any other questions or concerns regarding our paper or response.